# Effects of Kin Ball Initiation: Pre- and Post-Pandemic Impact on Palmar Muscle Strength, Endurance, and Coordination in Non-Athlete Participants

**DOI:** 10.3390/sports12060158

**Published:** 2024-06-06

**Authors:** Daniel Rosu, Ion-Sebastian Enache, Raul-Ioan Muntean, Valentina Stefanica

**Affiliations:** 1Department of Physical Education and Sport, Faculty of Sciences, Physical Education and Informatics, National University of Science and Technology Politehnica Bucharest, Pitesti University Center, 110040 Pitesti, Romania; daniel.rosu@upb.ro (D.R.); ion.enache1109@upb.ro (I.-S.E.); valentina.stefanica@upb.ro (V.S.); 2Department of Physical Education and Sport, Faculty of Law and Social Sciences, University “1 Decembrie 1918” of Alba Iulia, 510009 Alba Iulia, Romania

**Keywords:** Kin Ball initiation, physical activity constraints, hand strength, endurance, motor control

## Abstract

The aim of our research is to introduce Kin Ball for the first time in Romania and assess its impact on the motor capacities of practitioners, particularly focusing on its potential contribution to developing motor skills in young students within the academic sphere, despite the challenges posed by the COVID-19 pandemic. Design: A retrospective, case–control study with a focus on four distinct research groups. Setting: The research was conducted in a camp setting, situated in the mountainous region of Bughea de Sus, Romania. Participants: The study included 94 students, comprising 54 boys and 40 girls, with an overall average age of 20.85 ± 2 years. All participants were classified as non-athletes based on their level of sports practice. Interventions: The Kin Ball initiation program, a key component of the study, comprised 20 training sessions conducted in 10-day stages, systematically evaluating participants’ motor skills. Main outcome measures: The study assessed palmar muscle strength, endurance, and co-ordination function. Results: Statistical analyses, including the Kolmogorov–Smirnov test, revealed notable differences in the initiation process and significant variations (*p* < 0.05) in all measurements performed in 2022. In 2019, the tests recorded both significant and statistically insignificant differences, as indicated by the calculation of Cohen’s d indicator. Conclusions: The study underscored the influence of reduced movement during the pandemic on motor skills and highlighted Kin Ball’s potential as an alternative sport within physical education curricula. Despite lower baseline motor capacities observed in 2022, participants exhibited significant progress, emphasizing Kin Ball’s capacity to improve physical skills. These findings underscore the importance of alternative sports such as Kin Ball in fostering holistic personal development and mitigating the impact of pandemic-induced disruptions on motor skills.

## 1. Introduction

Recreational physical exercises are widely acknowledged for their substantial contributions to enhancing both the physical and mental well-being of individuals [1,2]. Physical education teachers often employ traditional methods rooted in cultural practices to promote physical activities among students [3]. While these approaches have proven valuable, there is growing recognition of the complementary benefits of diverse intervention programs, aiming to boost motivation, participation, and the development of motor skills in the youth generation [4].

The intervention programs encompass a broad spectrum of activities, including diversified sports, adventure programs, and alternative approaches utilizing a common tool—the ball [5,6,7]. The multifaceted use of the ball in various forms not only promotes fun and goodwill but also fosters the development of individual and collective physical and mental capacities. Ball games, often referred to as sports games, play a significant role in shaping group processes and phenomena, such as social facilitation, interpersonal relationships, communication, cohesion, and combating deindividuation [8,9,10].

Throughout history, the ball has captivated people across different cultures, evolving from ancient practices to contemporary sports such as football, basketball, handball, or volleyball, which are enjoyed recreationally, educationally, and performatively [11]. Motor activities involving the ball induce feelings of joy and positive mood, triggering the release of endorphins and enhancing the overall functions of the body [12,13]. While commonly associated with playful activities, the ball takes diverse forms in various bodily activities, including gymnastics, agonistic sports, recreational games, or compensatory sports activities [14,15,16].

In contemporary sports games, the ball serves as the central object around which sports activities are organized, acting as a means of expressing mastery and individual skills [17]. The ball’s diverse uses extend to artistic gymnastics, golf, bowling, and training in compensatory activities, highlighting its versatility in developing qualities and motor skills through methods like using weighted balls, target throwing, or distance throwing [18,19].

Despite the inherent benefits of ball-related activities, a concerning trend has emerged in physical education and sports, where educators focus excessively on the performance aspects of sports games rather than their educational facets [20]. This misdirection compromises the attainment of educational ideals, including cognitive development and personality formation among students [21].

The ball, varying in size, weight, and consistency based on specific regulations, remains pivotal in the development of contemporary sports, continually evolving in its manipulation forms [22,23]. The persistent effort to diversify the use of the ball has given rise to new forms of sports activities, often categorized as alternative sports [20]. The didactic and educational objectives of incorporating alternative sports in physical education activities encompass motor development, intellectual growth, co-operation, co-education, collaboration, commitment, and value formation [24]. Kin Ball, as a unique game using the ball in a special way, exhibits distinctive features in terms of size, weight, and utilization [25,26].

Kin Ball, an alternative sport of Canadian origin established in 1986 and regulated by the International Kin Ball Sport Federation, stands out for its collaborative gameplay and dynamic utilization of the giant ball [27]. Played in three teams of four players, Kin Ball prohibits the massive ball (1.2 m) from touching the ground, encouraging teamwork and collaborative spirit among teammates [28,29]. Despite its global popularity and positive effects on motivation, particularly among less active players, Kin Ball remains underexplored in scientific research, with limited studies focusing on standard conditions and initiation-learning methods [13,14,15,16,17,18,19,20,21,22,23,24,25,26,27,28,29,30].

The present research seeks to contribute to the existing body of knowledge by introducing Kin Ball for the first time in Romania, simultaneously measuring the effects on the motor capacities of practitioners. The research aims to address the following objectives:To measure and analyze the impact of various motor capacity components (palmar muscle strength, endurance, and co-ordination) in students engaging in Kin Ball initiation;To assess and compare changes in measured motor skills among participants before and after the acute phase of the COVID-19 pandemic, which imposed constraints on physical activity;To examine the educational significance of integrating Kin Ball into the physical education curricula in Romania, particularly in terms of its potential contribution to developing motor skills in young students and non-athletes.

By addressing these objectives, the research seeks to deepen our understanding of the initiation process in Kin Ball and its broader implications for motor skill development among students, with a particular focus on the challenges posed by the COVID-19 pandemic.

## 2. Materials and Methods

### 2.1. Study Design

We utilized a two-group repeated-measures experimental design to assess how participating in the Kin Ball initiation program affected various aspects of participants’ physical capabilities.

Participants for the two experimental groups were recruited from students enrolled in either the 2019 or 2022 editions of the extracurricular project “Freshman Camp”, organized by the University of Pitesti in Bughea de Sus, Arges, Romania. Recruitment was conducted during Physical Education and Sports classes as part of the students’ higher education curriculum. We did not provide any incentives to boost participation. We employed a conventional sampling approach, given that participants were university students and non-athletes with no prior experience in Kin Ball [31]. Additionally, we considered factors like age and gender to ensure the representativeness of our results.

Certain exclusion criteria were applied during participant selection. Students who did not provide written consent were excluded, as were those who had previously engaged in similar physical exercise programs to minimize the potential influence of prior experiences.

The research protocol involved a comprehensive approach, including 20 introductory Kin Ball training sessions spread over a 10-day period. We conducted three initial motor tests to assess participants’ palmar strength, running endurance, and general co-ordination.

It is essential to note that our study design accommodated the impact of the global COVID-19 pandemic, which began in March 2020. To capture the effects of pandemic-related constraints on physical activity, Group 1 underwent experimental procedures in September 2019, before the pandemic outbreak, while Group 2 conducted the experiment in September 2022, after the acute phase had concluded. This temporal distinction represents a significant contextual variation between the two cohorts.

During the initiation process into Kin Ball, participants engaged in a structured series of activities aimed not only at skill acquisition and development but also at fostering co-operation, healthy aspects of competition, and teamwork [32]. These activities included explanations, practical demonstrations, warm-up exercises, and hands-on practice of Kin ball’s distinctive elements. Moreover, technical procedures and organized games were incorporated to encourage fair play, tactical intelligence, and camaraderie among participants. The focus was not solely on achieving physical performance but also on promoting holistic personal development through engaging in the sport [33,34] (Figure 1).

Data collection occurred both before and after the intervention, with pre- and post-intervention tests conducted outdoors on the sports field of the mountain camp. Tests were conducted at the same time of day (between 5:00 p.m. and 6:30 p.m.) and under similar environmental conditions (temperature: 21–25 °C; humidity: 40–50%; and wind speed: ≤2 m/s) to ensure consistency. These assessments were administered by the same individuals from the project management team to maintain reliability and uniformity in the evaluation process.

### 2.2. Participants

Ninety-four young Romanian students participated in one of the two editions of the extracurricular project “Freshman Camp”. The study was conducted in accordance with the Declaration of Helsinki and approved by the Ethics Committee of Doctoral School of Physical Education and Sport Science of University of Pitesti (04/21.07.2019).

Table 1 presents the demographic characteristics and sports practice level of participants categorized by gender and research group. In Group 1, conducted in 2019, there were 24 male participants (54.5% of the total) with an average age of 21.5 ± 2.1 years and 20 female participants (45.5% of the total) with an average age of 20.8 ± 2.6 years. In Group 2, conducted in 2022, there were 29 male participants (58% of the total) with an average age of 20.4 ± 1.1 years and 21 female participants (42% of the total) with an average age of 20.6 ± 1.8 years. All participants were classified as non-athletes.

### 2.3. Data Collection Tools

The selection of specific tests and assessments, such as the Matorin Test for general body co-ordination and balance, the dynamometer for assessing palmar muscle strength, and the adaptation of the Cooper Test for endurance measurement over a 2 min interval, was guided by several factors. Firstly, these tests have been widely utilized and validated in previous research studies, demonstrating their reliability and effectiveness in assessing the desired motor capacities. Dynamometer-based evaluations for measuring muscle strength have been widely employed in studies assessing physical performance in various sports [35,36]. The Cooper Test has been validated through studies exploring cardiovascular fitness across diverse populations [37]. In our case, we adapted the Cooper test to correspond with our recreational objectives rather than emphasizing competition. The Matorin Test, assessing co-ordination and balance, has been recognized and applied in studies examining motor skills and co-ordination in sports contexts [38]. Furthermore, the chosen tests align with the nature of extracurricular activities and leisure pursuits for students, ensuring ease of application and maintaining participant engagement throughout the evaluation process. Additionally, the conditions of the camp constrained us in transporting modern research equipment.

#### 2.3.1. Palmar Muscle Strength

##### Palmar Muscle Strength Assessment (Using a Dynamometer)

During the palmar muscle strength assessment using a dynamometer, participants are required to exert maximum force by squeezing the dynamometer with the muscles of their hand. This action involves a gripping motion where participants firmly grasp the dynamometer and exert pressure with their fingers and palm. The evaluator closely observes this movement to ensure that participants are applying maximum force. The force exerted by the participant is then measured and recorded by the evaluator using the readings displayed on the dynamometer [39].

#### 2.3.2. Endurance

During the Adapted Cooper Test for 2 min, participants engage in continuous running for the duration of the test. They are instructed to run at their maximum capacity, aiming to cover as much distance as possible within the given 2 min timeframe. An evaluator oversees the test, monitoring the participants’ time and recording the distance covered by each individual during the 2 min interval [40].

#### 2.3.3. General Body Co-Ordination and Balance

The Matorin Test involves participants engaging in a variety of movements to evaluate their co-ordination and balance capabilities. These movements encompass:Twisting: Participants are directed to rotate their torso in various directions while ensuring stability. This action serves to assess rotational co-ordination and the strength of their core muscles.Bending: Participants execute forward bends to gauge their flexibility and balance, all while altering their center of gravity. This movement specifically targets the assessment of lower body co-ordination and stability.Weight shifting: Participants are tasked with transferring their body weight from one foot to the other and maintaining balance on a single leg. This challenge further tests their co-ordination and stability, requiring precise control and adjustment of their body position. The evaluator closely observes the participants as they execute the test movements, noting their performance and measuring the degrees of body rotation according to the specific instructions provided for the Matorin Test. To ensure accuracy and reliability, three trials of the test are typically conducted on each side of the body. Following the completion of all trials, the evaluator records the participants’ best performance for each side of the body [41].

These tools facilitated the recording, processing, and aggregation of individualized data for each participant.

### 2.4. Methodological and Organizational Adjustments

The shift in education towards holistic personal development rather than just athletic ability highlights the importance of alternative games and sports, offering diverse opportunities for specialists. These tools, like Kin Ball, can be integrated into physical education curricula to promote various outcomes such as motor skills, cognitive growth, social cohesion, and ethical values. Beyond its social aspects, Kin Ball is effective in developing essential motor skills like co-ordination, balance, and spatial perception [20,21,22,23,24,25,26,27,28,29,30,31,32,33,34,35,36,37,38,39,40,41,42].

To harness these benefits effectively, a methodology anchored in creativity and participant-driven exploration is imperative. By prioritizing playful and inventive approaches, specialists can orchestrate meaningful learning experiences tailored to individual needs and preferences, thus maximizing the transformative potential of Kin Ball within educational contexts [43,44].

During the initial training sessions, a 65 cm gym ball, smaller than the standard 122 cm Omnikin ball [45], was employed to familiarize participants with fundamental positions and movements. To simplify the learning process, the first simulated match-meetings were conducted with two teams instead of three, as stipulated in the official rules of Kin Ball. Team colors were assigned based on available materials, ensuring a pragmatic approach to organizational considerations.

These methodological and organizational adjustments were informed by established pedagogical principles and the need for a gradual and effective introduction to the complexities of Kin Ball.

### 2.5. Procedure of the Intervention

#### Kin Ball Experimental Program

The Kin Ball intervention program, outlined in Table 2, was meticulously designed to align with the fundamental instructional and learning objectives inherent in Kin Ball practice, drawing inspiration from various scholarly sources [46,47]. The program consisted of two-hour training sessions conducted twice daily over a period of 10 days during both the 2019 and 2022 interventions.

This structured program aimed to cover a diverse range of Kin Ball activities, incorporating elements of teamwork, co-ordination, and skill development. The selection of activities was based on established pedagogical principles and the goal of ensuring a comprehensive and engaging learning experience for the participants. Comparable intervention programs have been utilized in previous studies examining the impact of Kin Ball on various motor skills and cognitive abilities [13,20,30].

### 2.6. Data Analysis

The data analysis revealed differences in the distributions of variables related to strength and endurance compared to those associated with general co-ordination and balance. Given that the strength and endurance assessments followed normal distributions, we opted to use the Kolmogorov–Smirnov test for statistical analysis [48]. This allowed us to employ the Paired Sample *t*-test to compare measurements before and after the intervention [49].

The choice of the Paired Sample *t*-test was based on its suitability for data that follow a normal distribution. However, for variables related to general co-ordination and balance, where the data did not follow a normal distribution, we utilized the Wilcoxon Paired-Z test. This test is better suited for non-normally distributed data [50].

To assess the magnitude of the intervention’s impact, effect size was calculated using Cohen’s standardized d scale [51]. Interpretations were categorized as null (0–0.19), low (0.20–0.49), moderate (0.50–0.79), or high (≥0.80). This facilitated measurement of the differences between data points, conducted separately for males (M) and females (F).

Statistical analysis was conducted using SPSS, version 23.0, following similar methodologies employed in comparable studies assessing the effectiveness of physical activity interventions on motor skills [52].

## 3. Results

Table 3 and Table 4 present the outcomes of the descriptive, univariate, and bivariate analyses.

Table 5 provides the effect size calculations (Cohen’s d) for various variables, differentiating between males (M) and females (F).

The statistical analysis revealed several assessments with statistically significant differences (*p* < 0.05), indicating the effectiveness of the prescribed Kin Ball program. Specifically:Palmar muscle strength increased significantly in boys in 2019 [t (24) = 2.519; *p* < 0.05], girls in 2019 [t (20) = 4.224; *p* < 0.01], boys in 2022 [t (29) = 7.388; *p* < 0.01], and girls in 2022 [t (21) = 9.663; *p* < 0.01];Endurance demonstrated significant improvement in boys in 2019 [t (24) = 2.635; *p* < 0.05], boys in 2022 [t (29) = 6.027; *p* < 0.01], and girls in 2022 [t (21) = 5.630; *p* < 0.01];General body co-ordination and balance exhibited significant enhancements in girls in 2019 [z (20) = 2.521; *p* < 0.05], boys in 2022 [z (29) = 2.041; *p* < 0.05], and girls in 2022 [z (21) = 2.831; *p* < 0.01].

Additionally, there was an increase in average performance in certain assessments, although not statistically significant (*p* > 0.05):Endurance in girls in 2019 [t (20) = 1.073; *p* > 0.05];General body co-ordination and balance in boys in 2019 [z (24) = 1.131; *p* > 0.05].

## 4. Discussion

This study aimed to assess the impact of Kin Ball initiation on various motor capacity components, such as palmar muscle strength, endurance, and co-ordination, among students. Additionally, it aimed to compare changes in measured motor skills before and after the COVID-19 pandemic, which imposed constraints on physical activity. Lastly, it sought to explore the educational significance of integrating Kin Ball into Romanian physical education curricula, particularly its potential contribution to developing motor skills in young non-athlete students.

The findings of this study revealed a significant decrease in pre-intervention motor performance levels in 2022 compared to 2019 across all measured components. This decline suggests that Romanian students, faced with legal mandates and prolonged periods of online schooling during the COVID-19 pandemic, experienced notable impacts on their motor skills due to reduced physical activity opportunities.

Table 3 and Table 4 illustrate the substantial improvement in various motor capacity aspects among young non-athletes following their engagement in Kin Ball intervention. Notably, enhancements were observed in palmar muscle strength (PMS), endurance (END), and co-ordination–balance (CB) measures. These improvements were consistent across gender groups and time periods, indicating the efficacy of Kin Ball programs in enhancing physical fitness among young individuals.

While many improvements were statistically significant, it is crucial to note that not all reached conventional thresholds (*p* < 0.05), highlighting the importance of considering effect sizes alongside statistical significance. The use of both parametric and non-parametric tests ensured the validity of the statistical analysis, confirming the effectiveness of the Kin Ball intervention.

The notable improvement observed in 2022 can be attributed to the reduced initial motor capabilities caused by pandemic-related isolation [53,54]. Despite starting with lower motor levels compared to 2019, students showed easier progress with physical exercises and Kin Ball practice.

The findings underscore the positive impact of Kin Ball programs on motor skill development among non-athlete students.

In the context of existing research, our results align with the work of other specialists who have examined the impact of various interventions on co-ordination, palmar muscle strength, and endurance. For instance, Tulbure et al. [55] aimed to objectively assess the development of motor quality indices, specifically strength, in 18 students using the circuit method. The findings indicated that, contrary to the initial hypothesis, while there was a slight improvement in the experimental group using the circuit method, it was not conclusively proven to be more successful than the method involving dynamic games for strength development in eighth graders. Podstawski et al. undertook a study to evaluate the relationship between various physical activities chosen by 337 first-year full-time male students and their motor abilities, among other factors. The findings indicated associations between the selection of physical activities and motor fitness, highlighting noteworthy enhancements in motor abilities for students participating in activities such as general Physical Education, martial arts, jogging with sauna, and volleyball [56]. Hollerbach et al. investigated the impact of physical activity classes, specifically traditional weight training and CrossFit, on various fitness parameters, including grip strength and general endurance, in 85 healthy college students. While the study did not find significant differences between the two groups in terms of palmar muscle strength improvement, both classes were effective in enhancing endurance, emphasizing the potential of activity classes to contribute to students’ physical fitness [57]. Vaida conducted a study with the primary aim of assessing the development of co-ordinative capacity/balance in students, particularly focusing on demonstrating the potential improvement of balance through specific means of physical education. The research involved testing 56 students (36 girls and 20 boys) using the Bass test. The results revealed relatively small differences between boys and girls, emphasizing that enhancing co-ordination skills, specifically balance, is achievable in adults through dedicated physical education programs, with the potential to accelerate the overall development of motor skills [58].

The study’s outcomes support existing research on Kin Ball, emphasizing its benefits for specialists in diversifying co-operative, social, and cognitive activities.

In the transition towards an educational focus more geared towards holistic personal development rather than solely athletic achievement, alternative games and sports play a fundamental role due to the opportunities they offer educators. These can be utilized in physical education classes in a variety of ways to achieve diverse outcomes, whether it be motor development, intellectual growth, co-operation, co-education, collaboration, engagement, or values formation. Alongside fostering co-operation, healthy aspects of competition, and teamwork, Kin Ball also promotes and encourages other factors such as fair play, tactical intelligence, and camaraderie. Additionally, it facilitates the integration of less skilled individuals as it is a sport that is very easy to learn and, due to its novelty, all students start from a similar base. Moreover, since teams are mixed, gender equality is favored [59,60].

This approach recognizes that the journey towards physical development extends beyond mere technical proficiency, encompassing the enjoyment, challenges, and personal growth experienced throughout the process. It acknowledges the importance of engaging learners in diverse and meaningful experiences that not only enhance their physical abilities but also instill a lifelong appreciation for active living.

## 5. Limitations and Challenges

The study’s limitation is primarily due to the unavoidable influence of the global COVID-19 pandemic and legal mandates, which resulted in prolonged periods of online schooling for participants. This situation significantly limited opportunities for physical activity, affecting individual motor capacities and potentially introducing confounding variables into the study’s results.

Methodological limitations were acknowledged due to the implementation of streamlined testing procedures, strategically designed to minimize disruptions to the Freshman Camp programs in both 2019 and 2022.

The research instruments used include potential variability in participant technique and effort with the dynamometer, as well as challenges in standardizing and ensuring the reliability of the Matorin Test due to subjective evaluation. The accuracy of the Cooper test may be influenced by participant compliance and environmental factors.

Future research could concentrate on standardizing testing protocols, providing examiner training for consistency, and exploring alternative assessment tools tailored to Kin Ball-specific physical attributes. These efforts could improve research outcomes and enhance the overall quality of the study.

To enhance the learning process, methodological and organizational adjustments were made, such as using a 65 cm gymnastic ball in the initial training sessions and conducting simulated matches with two teams instead of three. While demonstrating the potential benefits, the study recognizes the unfamiliarity of Kin Ball to many physical education instructors due to its limited popularity and challenges associated with accessing specific materials required for practice, such as the 48-inch ball.

## 6. Conclusions

This study successfully achieved its proposed objectives, shedding light on the multifaceted impact of Kin Ball initiation on various motor capacity components among students. Through meticulous measurement and analysis, the research elucidated the significant enhancement of palmar muscle strength, endurance, and co-ordination among participants engaging in Kin Ball activities.

Furthermore, the study effectively compared changes in measured motor skills before and after the acute phase of the COVID-19 pandemic, providing valuable insights into the influence of pandemic-related constraints on physical activity. The observed decline in pre-intervention motor performance levels in 2022, compared to 2019, underscores the profound impact of the pandemic on participants’ motor skills.

Importantly, the research explored the educational significance of integrating Kin Ball into the physical education curricula in Romania, particularly emphasizing its potential contribution to developing motor skills in young non-athlete students. By fostering co-operation, gender equity, and ethical values, Kin Ball emerges as a valuable tool for promoting holistic personal development and inclusivity within physical education programs.

These findings contribute significantly to the ongoing discourse surrounding the educational benefits of alternative games and sports. They underscore the pivotal role of Kin Ball in not only improving physical fitness but also enhancing diverse outcomes and individuals’ overall well-being.

Looking ahead, future research endeavors can delve deeper into the long-term effects of Kin Ball practice and explore its broader implementation in educational settings to maximize its potential benefits. By continuing to explore and harness the potential of Kin Ball, educators and policymakers can further enrich physical education programs and promote the holistic development of students.

## Figures and Tables

**Figure 1 sports-12-00158-f001:**
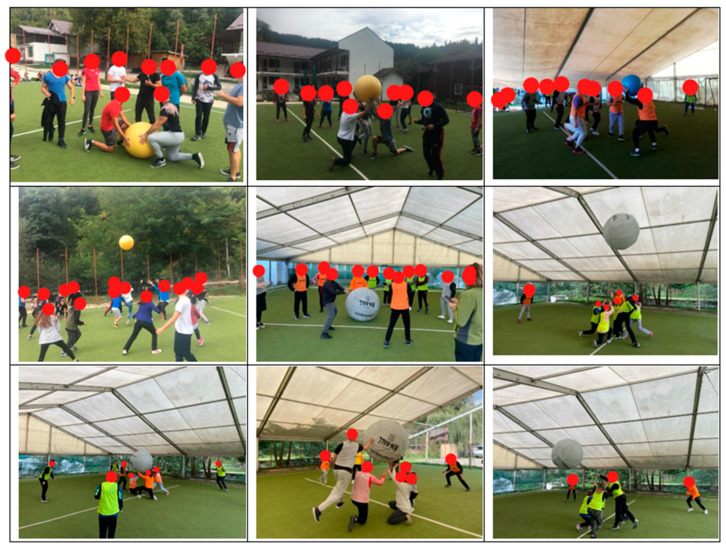
Poses of Kin Ball initiation and teaching (personal photo collection).

**Table 1 sports-12-00158-t001:** Demographic characteristics and sports practice level of participants.

Group and Year	Gender	Number	Percentage (%)	Average Age (Years)	Sports Practice Level
Group 1—year 2019	Male	24	54.5	21.5 ± 2.1	Non-athlete
Group 1—year 2019	Female	20	45.5	20.8 ± 2.6	Non-athlete
Group 2—year 2022	Male	29	58	20.4 ± 1.1	Non-athlete
Group 2—year 2022	Female	21	42	20.6 ± 1.8	Non-athlete

**Table 2 sports-12-00158-t002:** List of activities of the Kin Ball intervention program.

Characteristics			Sessions		
	1–4	5–8	9–12	13–16	17–20
Activity	Poison	Four corners	Indiana Jones	Four corners	Indiana Jones
	Tripod	Together	Tripod	Centipede	Four corners
	Indiana Jones	Kin Ball Omnikin	Poison	Ultimate rugby	Together
	Centipede	Ultimate rugby	Together	Together	Tripod
	Ultimate rugby	Match	Match	Match	Match

Note: Centipede—Participants lie on their backs, passing the ball; Tripod—Three participants hold the ball together; Indiana Jones—A participant runs in a circle, avoiding ball hits; Kin Ball Omnikin—Hitting the ball with both hands; Together—All teammates touch the ball simultaneously; Ultimate rugby—Passing a small ball without interception; Poison—Aiming the ball at other players; Four corners—Participants in four corners volley the ball to another corner; Match—Simulations of match scenarios.

**Table 3 sports-12-00158-t003:** Comparative results of the palmar muscle strength (PMS) and endurance (END) tests performed pre- and post-intervention with Kin Ball means, with normal distributions.

Variables	Group 1/2 M/F	No.	M	SD	Kolmogorov–Smirnov Test, *p* ≥ 0.05	Paired Sample *t* Test	sig	*p*
PMS pre-2019	1M	24	45.25	4.436	0.154	2.519	0.019	<0.05
PMS post 2019	1M	24	46.25	4.436	0.200 *			
PMS pre-2019	1F	20	22.45	2.856	0.200 *	4.224	0.001	<0.01
PMS post 2019	1F	20	27.10	3.626	0.200 *			
PMS pre-2022	2M	29	39.90	4.117	0.200 *	7.388	0.001	<0.01
PMS post 2022	2M	29	45.66	3.994	0.075			
PMS pre-2022	2F	21	20.38	2.500	0.200 *	9.663	0.001	<0.01
PMS post 2022	2F	21	31.57	4.632	0.050			
END pre-2019	1M	24	800.42	46.483	0.200 *	2.635	0.015	<0.05
END post 2019	1M	24	807.08	44.476	0.176			
END pre-2019	1F	20	652	28.023	0.197	1.073	0.297	>0.05
END post 2019	1F	20	654	29.272	0.200 *			
END pre-2022	2M	29	729.66	56.283	0.200 *	6.027	0.001	<0.01
END post 2022	2M	29	783.45	54.266	0.200 *			
END pre-2022	2F	21	641.43	35.396	0.054	5.630	0.001	<0.01
END post 2022	2F	21	690.00	32.249	0.052			

Note: palmar muscle strength (MF)—indicator of dynamometer; endurance (RES)—measuring in meters; male (M); female (F). * This is the lower bound of the true significance.

**Table 4 sports-12-00158-t004:** Comparative results of co-ordination–balance (CB) pre- and post-intervention with Kin Ball means, with asymmetric distributions.

Variables	Group 1/2 M/F	No.	M	SD	K–S * *p* ≤ 0.05	Wilcoxon Pair- Z test	sig	*p*
CB pre-2019	1M	24	344.17	33.221	0.000	1.131	0.258	>0.05
CB post 2019	1M	24	346.25	30.333	0.000			
CB pre-2019	1F	20	310.00	38.389	0.002	2.521	0.012	<0.05
CB post 2019	1F	20	326.00	39.256	0.001			
CB pre-2022	2M	29	344.14	31.681	0.000	2.041	0.041	<0.05
CB post 2022	2M	29	349.66	24.854	0.000			
CB pre-2022	2F	21	302.86	33.637	0.011	2.831	0.005	<0.01
CB post 2022	2F	21	319.52	37.212	0.036			

Note: co-ordination–balance (CB)—measuring in degree (0–360); male (M); female (F). * Kolmogorov–Smirnov test.

**Table 5 sports-12-00158-t005:** Effect size calculation (Cohen’s d).

Variables	M/F	t/z	gl	r^2^	d- Cohen	Explain
Muscle strength 2019	Male	2.519	23	0.216	1.05	high
Muscle strength 2019	Female	4.224	19	0.484	1.938	high
Muscle strength 2022	Male	7.388	28	0.661	2.792	high
Muscle strength 2022	Female	9.663	23	0.802	4.03	high
Endurance 2019	Male	2.635	23	0.232	1.099	high
Endurance 2019	Female	1.073	19	0.057	0.492	low
Endurance 2022	Male	6.027	28	0.565	2.278	high
Endurance 2022	Female	5.630	23	0.58	2.348	high
Coord–Balance 2019	Male	1.131	23	0.053	0.472	low
Coord–Balance 2019	Female	2.521	19	0.251	1.157	high
Coord–Balance 2022	Male	2.041	28	0.13	0.771	moderate
Coord–Balance 2022	Female	2.831	23	0.258	1.181	high

## Data Availability

Data available on request due to restrictions e.g., privacy or ethical. The data presented in this study are available on request from the corresponding author. The data are not publicly available due to confidentiality.

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
