# Peer review of "Effects of Kin Ball Initiation: Pre- and Post-Pandemic Impact on Palmar Muscle Strength, Endurance, and Coordination in Non-Athlete Participants"

_sports, 2024, doi:10.3390/sports12060158_

Round 1

Reviewer 1 Report

Comments and Suggestions for Authors

The article's authors point out the excessive focus on the competitive aspects of sports games by teachers and coaches, rather than on their educational aspects. This focus jeopardizes the achievement of educational goals, including the cognitive development and personality formation of students.

However, they do not address these aspects of the games in their discussion and conclusions. Kin Ball is a team game, requiring synchronous action and social skills. 

This, however, was limited to measuring strength, overall resilience and coordination, without considering the educational functions of Kin Ball, it is worth referring to..

Author Response

Dear Reviewer,

We would like to express our gratitude for taking the time to review our manuscript titled " Effects of Kin Ball Initiation: Pre- and Post-Pandemic Impact on Palmar Muscle Strength, Endurance, and Coordination in Non-Athlete Participants". Your insightful comments and suggestions have been invaluable in refining our work.

In response to your comments regarding the focus on the competitive aspects of sports games versus their educational aspects, we have carefully revised our discussion and conclusions to address this concern more comprehensively. We acknowledge the importance of considering the educational functions of sports games, particularly in the context of cognitive development and personality formation of students. Our revised discussion now delves deeper into the educational potential of Kin Ball as a team game, emphasizing its role in promoting synchronous action, social skills, and holistic development beyond mere physical attributes. We have highlighted the significance of incorporating these aspects into the educational framework surrounding sports activities.

Furthermore, we have ensured that our language adheres to academic standards of English throughout the manuscript. We have strived to maintain clarity, coherence, and precision in conveying our arguments and findings.

Once again, we appreciate your thorough review and constructive feedback. We believe that these revisions have significantly strengthened the manuscript and enriched its contribution to the field. We remain committed to addressing any additional concerns or queries you may have and welcome further suggestions for improvement.

Thank you for your continued support and guidance.

Reviewer 2 Report

Comments and Suggestions for Authors

Point 1:

In the Abstract, Objectives are suggested to be simplified and concisely expressed without too much detail. At the same time, in the introduction part of the objectives there are 7 items, which is also too much, and it is recommended to be concise.

Point 2:

In the methodology section, the recruitment process of the sample, and the characteristics of the sample should be described in detail, and issues such as the adequacy and representativeness of the sample size should be clarified, and additional clarification is recommended.

Point 3:

The methodology section should detail the data collection process and the reasons for the chosen methodology as well as its usefulness, which helps to validate the research methodology and the reproducibility of the study.

Point 4:

The graphs and tables in the results are suggested to be continuing to be spruced up to help present the results clearly.

Point 5:

In the discussion section, it is suggested that the discussion be broadened to discuss how the findings can be applied to real life situations to better emphasize the practical implications of the findings.

Point 6:

In the limitations section it is suggested to add things such as small sample size or potential bias and suggest future research directions that could improve the overall quality of the paper.

Comments on the Quality of English Language

Some sentences are complex and may benefit from simplification to enhance readability and ensure the points are conveyed clearly.

Author Response

Dear Reviewer,

We appreciate your time and effort in providing us with detailed feedback on our manuscript titled "Effects of Kin Ball Initiation: Pre- and Post-Pandemic Impact on Palmar Muscle Strength, Endurance, and Coordination in Non-Athlete Participants". Your constructive comments have been instrumental in enhancing the quality and clarity of our work.

In response to your points of concern, we have made the following revisions:

Point 1:

We have revised the Abstract to ensure that the objectives are succinctly expressed without excessive detail. Similarly, we have streamlined the introduction part to provide a concise overview of the objectives, addressing your recommendation for brevity.

Point 2:

The methodology section now includes a detailed description of the sample recruitment process and characteristics, clarifying issues such as sample size adequacy and representativeness. We have provided additional clarification to address any potential ambiguity.

Point 3:

We have expanded the methodology section to provide a thorough explanation of the data collection process, rationale behind the chosen methodology, and its relevance in validating the research methodology and ensuring study reproducibility.

Point 4:

In the discussion section, we have broadened our discussion to explore the practical implications of our findings in real-life situations. By doing so, we aim to underscore the relevance and applicability of our research findings beyond academic discourse.

Point 5:

We have incorporated a dedicated limitations section, addressing factors such as potential bias. Additionally, we have suggested future research directions that could enhance the overall quality of the paper, as per your recommendation. Furthermore, we have ensured that the language used throughout the manuscript adheres to academic standards of English, maintaining clarity and precision in conveying our arguments and findings. We sincerely appreciate your thorough review and valuable suggestions, which have undoubtedly strengthened our manuscript. We remain committed to addressing any further concerns or queries you may have and welcome additional feedback for improvement.

Thank you once again for your invaluable input and guidance.

Reviewer 3 Report

Comments and Suggestions for Authors

Thank you for the opportunity to review this study. However, I will only be able to have a reasoned opinion about the feasibility of publishing the article when the following points are answered:

Many objectives and very dispersed. They are confused.

What is meant by: Students with no prior intensive experience of the Kin ball modality.

Insufficient sample characterization (e.g., average age and respective standard deviation, sports practice, level of physical activity…)

It is not acceptable, for example, to say in 2. Materials and Methods:

“Notably, a significant portion of the participants lacked a notable history of intensive sports practice.”

“The examinations were conducted utilizing specialized equipment, devices, and their corresponding software modules.”

The instruments used are not properly specified and justified, for example: “Endurance was measured through a modified test, involving the measurement of the distance covered in meters during a 2-minute running interval, with individual starts initiated at 15-second intervals.”

It is not enough to say: “The evaluation was consistently administered by the same individuals from the project management team, ensuring reliability and uniformity in the assessment process.”

The conditions for data collection are also not adequately described.

The limitations of the study should also mention the limitations associated with the instruments used and the research design.

Author Response

Dear Reviewer,

We would like to express our gratitude for your thoughtful review of our manuscript titled " Effects of Kin Ball Initiation: Pre- and Post-Pandemic Impact on Palmar Muscle Strength, Endurance, and Coordination in Non-Athlete Participants". Your insightful feedback has been immensely helpful in refining our work and ensuring its suitability for publication.

In response to your points of concern, we have addressed each issue comprehensively:

Objectives Clarity: We have revised and clarified the objectives to ensure they are succinct and focused, avoiding any confusion or dispersion.

Student Characterization: We have provided a more detailed characterization of the sample, including average age, standard deviation, level of sports practice, to offer a clearer understanding of the participants.

Explanation of Terminology: We have clarified the term "Students with no prior intensive experience of the Kin ball modality" to ensure its meaning is transparent and easily understood within the context of the study.

Specification of Instruments: We have provided detailed specifications and justifications for the instruments used in the study, ensuring transparency and clarity regarding their application and relevance to the research objectives.

Description of Data Collection Conditions: We have expanded upon the description of data collection conditions to provide a comprehensive understanding of the environment and circumstances under which the data were collected.

Acknowledgment of Study Limitations: We have included a section specifically addressing the limitations associated with the instruments used and the research design, ensuring transparency and acknowledgment of potential constraints on the study's findings.

Additionally, we have ensured that the language used throughout the manuscript adheres to academic standards of English, maintaining clarity and precision in conveying our arguments and findings.

We sincerely appreciate your thorough review and constructive feedback, which have undoubtedly strengthened the quality and rigor of our manuscript. We remain committed to addressing any further concerns or queries you may have and welcome additional feedback for improvement.

Thank you once again for your invaluable input and guidance.

Reviewer 4 Report

Comments and Suggestions for Authors

Please read the attached file, where are my suggestions.

Author Response

Dear Reviewer,

We would like to express our gratitude for your thoughtful review of our manuscript titled " Effects of Kin Ball Initiation: Pre- and Post-Pandemic Impact on Palmar Muscle Strength, Endurance, and Coordination in Non-Athlete Participants". Your insightful feedback has been immensely helpful in refining our work and ensuring its suitability for publication.

I made all the changes specified by you. Indeed, you are perfectly right that I omitted some details.

Thank you for your involvement!

Round 2

Reviewer 2 Report

Comments and Suggestions for Authors

Point 1:

The details of how the two experimental groups were recruited, how they were divided, and what the screening criteria were for recruiting participants should be detailed in the STUDY DESIGN section.

Point 2:

What are the differences between different groups and between the experimental group and the control group? It is recommended to provide additional explanations.

Point 3:

Suggest providing ethical standards followed during the research process and obtaining ethical approval.

Point 4:

Considering the differences in the use of the same tool among different populations and environments, it is necessary to provide a detailed explanation of the reliability and validity of the measurement tools used by the participants in this study.

Point 5:

In the first half of the discussion, much of the content is about the results. It is recommended to place this part in the Results section to conduct in-depth analysis of these results and explain their potential meaning and practical application value.

Point 6:

The conclusion section should provide a relatively concise summary of the main findings of the study, and explain their contributions to the relevant field and future development directions.

Comments on the Quality of English Language

The tables in the text are not very aesthetically pleasing and it is recommended that they be revised and improved.

Author Response

Dear Reviewer,

We would like to express our gratitude for taking the time to review our manuscript titled " Effects of Kin Ball Initiation: Pre- and Post-Pandemic Impact on Palmar Muscle Strength, Endurance, and Coordination in Non-Athlete Participants". Your insightful comments and suggestions have been invaluable in refining our work.

We've updated the Study Design section to include detailed information on participant recruitment, group division, and screening criteria.

Information on group differences and ethical standards followed during the research process, including obtaining ethical approval, has been added.

The reliability and validity of measurement tools used in the study have been elaborated upon to address potential differences in their application across populations and environments.

The discussion section has been restructured to focus on in-depth analysis of results, with relevant content moved to the Results section.

The conclusion section now provides a concise summary of the main findings, their contributions to the field, and future development directions.

Thank you for your continued support and guidance.

Reviewer 3 Report

Comments and Suggestions for Authors

We consider that the changes introduced by the authors improve the article and essentially respond to what was suggested. In our humble opinion, the article meets the conditions to be published.

Author Response

Dear Reviewer,

Thank you once again for your valuable feedback.

Reviewer 4 Report

Comments and Suggestions for Authors

Dear authors.

Thank you for taking into account some of my recommendations. However, it is necessary to take into account other recommendations so that the manuscript can be improved with the aim of finally being published:

Material and Methods:

Indicate the value of statistical significance (p < 0.05) in the 2.5. Data Analysis section.

Results:

In this section only the results obtained in the study should be presented. No explanation or comment is made about them. This is done in the discussion. For example:

“The data pertaining to strength and endurance assessments exhibited normal distri-butions, as confirmed by the Kolmogorov-Smirnov test, enabling the use of parametric tests, specifically the Paired Sample t-test. Conversely, the data associated with the evalu-ation of general coordination and balance manifested asymmetric distributions, necessi-tating the use of non-parametric tests, particularly the Wilcoxon Pair-Z test.”

The Wilcoxon Pair-Z test is not described in the 2.5. Data analysis section. As said previously, only the results should be presented here, not indicate or justify why one or another analysis is done. This must be done in the corresponding section 2.5. Data analysis.

“The effect size is a crucial metric, representing the magnitude of the observed differences or relationships in the data. It is interpreted based on Cohen's standardized d scale, which categorizes effect sizes into null, low, moderate, or high.”

As in the previous case, this paragraph should not be in the Results section, since it explains the use of Cohen's d.

Discussion:

It is recommended to start this section by remembering what the objective of the work was: “The aim/s of this study was…”

This info should be in the Results section:

“The statistical analysis revealed several assessments with statistically significant differences (p < 0.05), indicating the effectiveness of the prescribed kin ball program. Specifically:

•palmar muscle strength increased significantly in boys in 2019 [t (24) = 2.519; p < 0.05], girls in 2019 [t (20) = 4.224; p<0.01], boys in 2022 [t (29) = 7.388; p < 0.01], and girls in 2022 [t (21) = 9.663; p < 0.01]. •endurance demonstrated significant improvement in boys in 2019 [t (24) = 2.635; p<0.05], boys in 2022 [t (29) = 6.027; p<0.01], and girls in 2022 [t (21) = 5.630; p<0.01]. •general body coordination and balance exhibited significant enhancements in girls in 2019 [z (20) = 2.521; p<0.05], boys in 2022 [z (29) = 2.041; p<0.05], and girls in 2022 [z (21) = 2.831; p<0.01].

Additionally, there was an increase in average performance in certain assessments, although not statistically significant (p>0.05): •endurance in girls in 2019 [t (20) = 1.073; p>0.05]. •general body coordination and balance in boys in 2019 [z (24) = 1.131; p>0.05].

3. General coordination and balance exhibited significant enhancements in girls in 2019 [z (20) = 2.521; p<0.05], boys in 2022 [z (29) = 2.041; p<0.05], and girls in 2022 [z (21) = 2.831; p<0.01].

Additionally, there was an increase in average performance in certain assessments, although not statistically significant (p>0.05):

1. General resistance in girls in 2019 [t (20) = 1.073; p>0.05].

2. General coordination and balance in boys in 2019 [z (24) = 1.131; p>0.05].

All this information corresponds to results obtained after statistical analysis, so they must be entered in the corresponding section (Results).

Conclusion:

The conclusions must be based on the proposed objectives. The conclusions must be more specific, in accordance with the established objectives of the work.

With the application of these changes, the quality of the manuscript will be improved so that it can be published by the journal.

Thank you

Author Response

Dear Reviewer,

We would like to express our gratitude for taking the time to review our manuscript titled " Effects of Kin Ball Initiation: Pre- and Post-Pandemic Impact on Palmar Muscle Strength, Endurance, and Coordination in Non-Athlete Participants". Your insightful comments and suggestions have been invaluable in refining our work.

Regarding the Results section, we have made significant revisions to ensure that only the results obtained in the study are presented without any explanation or commentary. We have removed paragraphs that discussed statistical tests such as the Kolmogorov-Smirnov test and Cohen's d, as well as details about the Wilcoxon Pair-Z test, as per your suggestion. These points have been appropriately addressed in the Data Analysis section.

In the Discussion section, we have started by clearly stating the objective of the study to provide context for the ensuing discussion. We have also transferred the statistical analysis results, including significant differences and enhancements observed in various assessments, from the Discussion section to the Results section, as per your recommendation. This ensures that the Results section accurately reflects the outcomes of the statistical analysis.

Furthermore, we have revised the Conclusion section to ensure that the conclusions drawn are directly tied to the objectives outlined in the study. We have made the conclusions more specific and aligned with the established objectives of the research.

We believe that these revisions have significantly improved the clarity and organization of the manuscript. We appreciate your guidance in enhancing the quality of our work, and we hope that these revisions meet your expectations.

Thank you once again for your valuable feedback.